# FRATERNAL DROPOUT

**Konrad Żołna**[1,2,*]**, Devansh Arpit**[2]**, Dendi Suhubdy**[2] **& Yoshua Bengio**[2,3]

[1]Jagiellonian University
[2]MILA, Université de Montréal
[3]CIFAR Senior Fellow

## ABSTRACT

Recurrent neural networks (RNNs) form an important class of architectures among neural networks useful for language modeling and sequential prediction. However, optimizing RNNs is known to be harder compared to feed-forward neural networks. A number of techniques have been proposed in literature to address this problem. In this paper we propose a simple technique called *fraternal dropout* that takes advantage of dropout to achieve this goal. Specifically, we propose to train two identical copies of an RNN (that share parameters) with different dropout masks while minimizing the difference between their (pre-softmax) predictions. In this way our regularization encourages the representations of RNNs to be invariant to dropout mask, thus being robust. We show that our regularization term is upper bounded by the expectation-linear dropout objective which has been shown to address the gap due to the difference between the train and inference phases of dropout. We evaluate our model and achieve state-of-the-art results in sequence modeling tasks on two benchmark datasets – Penn Treebank and Wikitext-2. We also show that our approach leads to performance improvement by a significant margin in image captioning (Microsoft COCO) and semi-supervised (CIFAR-10) tasks.

## 1 INTRODUCTION

Recurrent neural networks (RNNs) like long short-term memory (LSTM; Hochreiter & Schmidhuber (1997)) networks and gated recurrent unit (GRU; Chung et al. (2014)) are popular architectures for sequence modeling tasks like language generation, translation, speech synthesis, and machine comprehension. However, they are harder to optimize compared to feed-forward networks due to challenges like variable length input sequences, repeated application of the same transition operator at each time step, and largely-dense embedding matrix that depends on the vocabulary size. Due to these optimization challenges in RNNs, the application of batch normalization and its variants (layer normalization, recurrent batch normalization, recurrent normalization propagation) have not been as successful as their counterparts in feed-forward networks (Laurent et al., 2016), although they do considerably provide performance gains. Similarly, naive application of dropout (Srivastava et al., 2014) has been shown to be ineffective in RNNs (Zaremba et al., 2014). Therefore, regularization techniques for RNNs is an active area of research.

To address these challenges, Zaremba et al. (2014) proposed to apply dropout only to the non-recurrent connections in multi-layer RNNs. Variational dropout (Gal & Ghahramani (2016)) uses the same dropout mask throughout a sequence during training. DropConnect (Wan et al., 2013) applies the dropout operation on the weight matrices. Zoneout (Krueger et al. (2016)), in a similar spirit with dropout, randomly chooses to use the previous time step hidden state instead of using the current one. Similarly as a substitute for batch normalization, layer normalization normalizes the hidden units within each sample to have zero mean and unit standard deviation. Recurrent batch normalization applies batch normalization but with unshared mini-batch statistics for each time step (Cooijmans et al., 2016).

---

*konrad.zolna@gmail.com

Merity et al. (2017a) and Merity et al. (2017b) on the other hand show that activity regularization (AR) and temporal activation regularization (TAR)[1] are also effective methods for regularizing LSTMs. Another more recent way of regularizing RNNs, that is similar in spirit to the approach we take, involves minimizing the difference between the hidden states of the original and the auxiliary network Serdyuk et al. (2017).

In this paper we propose a simple regularization based on dropout that we call *fraternal dropout*, where we minimize an equally weighted sum of prediction losses from two identical copies of the same LSTM with different dropout masks, and add as a regularization the $\ell^2$ difference between the predictions (pre-softmax) of the two networks. We analytically show that our regularization objective is equivalent to minimizing the variance in predictions from different i.i.d. dropout masks; thus encouraging the predictions to be invariant to dropout masks. We also discuss how our regularization is related to expectation linear dropout Ma et al. (2016), Π-model Laine & Aila (2016) and activity regularization Merity et al. (2017b), and empirically show that our method provides non-trivial gains over these related methods which we explain furthermore in our ablation study (Section 5).

## 2 FRATERNAL DROPOUT

Dropout is a powerful regularization for neural networks. It is usually more effective on densely connected layers because they suffer more from overfitting compared with convolution layers where the parameters are shared. For this reason dropout is an important regularization for RNNs. However, dropout has a gap between its training and inference phase since the latter phase assumes linear activations to correct for the factor by which the expected value of each activation would be different Ma et al. (2016). In addition, the prediction of models with dropout generally vary with different dropout mask. However, the desirable property in such cases would be to have final predictions be invariant to dropout masks.

As such, the idea behind *fraternal dropout* is to train a neural network model in a way that encourages the variance in predictions under different dropout masks to be as small as possible. Specifically, consider we have an RNN model denoted by $\mathcal{M}(\theta)$ that takes as input $\mathbf{X}$, where $\theta$ denotes the model parameters. Let $\mathbf{p}^t(\mathbf{z}^t, s_i^t; \theta) \in \mathbb{R}^m$ be the prediction of the model for input sample $\mathbf{X}$ at time $t$, for dropout mask $s_i^t$ and current input $\mathbf{z}^t$, where $\mathbf{z}^t$ is a function of $\mathbf{X}$ and the hidden states corresponding to the previous time steps. Similarly, let $\ell^t(\mathbf{p}^t(\mathbf{z}^t, s_i^t; \theta), \mathbf{Y})$ be the corresponding $t^{th}$ time step loss value for the overall input-target sample pair $(\mathbf{X}, \mathbf{Y})$.

Then in *fraternal dropout*, we simultaneously feed-forward the input sample $\mathbf{X}$ through two identical copies of the RNN that share the same parameters $\theta$ but with different dropout masks $s_i^t$ and $s_j^t$ at each time step $t$. This yields two loss values at each time step $t$ given by $\ell^t(\mathbf{p}^t(\mathbf{z}^t, s_i^t; \theta), \mathbf{Y})$, and $\ell^t(\mathbf{p}^t(\mathbf{z}^t, s_j^t; \theta), \mathbf{Y})$. Then the overall loss function of *fraternal dropout* is given by,

$$\ell_{FD}(\mathbf{X}, \mathbf{Y}) = \sum_{t=1}^{T} \frac{1}{2} \left( \ell^t(\mathbf{p}^t(\mathbf{z}^t, s_i^t; \theta), \mathbf{Y}) + \ell^t(\mathbf{p}^t(\mathbf{z}^t, s_j^t; \theta), \mathbf{Y}) \right) + \frac{\kappa}{mT} \sum_{t=1}^{T} \mathcal{R}_{FD}(\mathbf{z}^t; \theta) \tag{1}$$

where $\kappa$ is the regularization coefficient, $m$ is the dimensions of $\mathbf{p}^t(\mathbf{z}^t, s_i^t; \theta)$ and $\mathcal{R}_{FD}(\mathbf{z}^t; \theta)$ is the *fraternal dropout* regularization given by,

$$\mathcal{R}_{FD}(\mathbf{z}^t; \theta) := \mathbb{E}_{s_i^t, s_j^t} \left[ \| \mathbf{p}^t(\mathbf{z}^t, s_i^t; \theta) - \mathbf{p}^t(\mathbf{z}^t, s_j^t; \theta) \|_2^2 \right]. \tag{2}$$

We use Monte Carlo sampling to approximate $\mathcal{R}_{FD}(\mathbf{z}^t; \theta)$ where $\mathbf{p}^t(\mathbf{z}^t, s_i^t; \theta)$ and $\mathbf{p}^t(\mathbf{z}^t, s_j^t; \theta)$ are the same as the one used to calculate $\ell^t$ values. Hence, the additional computation is negligible.

We note that the regularization term of our objective is equivalent to minimizing the variance in the prediction function with different dropout masks as shown below (proof in the appendix).

---

[1] TAR and Zoneout are similar in their motivations because both leads to adjacent time step hidden states to be close on average.

**Remark 1.** *Let $s_i^t$ and $s_j^t$ be i.i.d. dropout masks and $\mathbf{p}^t(\mathbf{z}^t, s_i^t; \theta) \in \mathbb{R}^m$ be the prediction function as described above. Then,*

$$\mathcal{R}_{FD}(\mathbf{z}^t; \theta) = \mathbb{E}_{s_i^t, s_j^t}\left[\|\mathbf{p}^t(\mathbf{z}^t, s_i^t; \theta) - \mathbf{p}^t(\mathbf{z}^t, s_j^t; \theta)\|_2^2\right] = 2\sum_{q=1}^{m} \text{var}_{s_i^t}(p_q^t(\mathbf{z}^t, s_i^t; \theta)). \quad (3)$$

Note that a generalization of our approach would be to minimize the difference between the predictions of the two networks with different data/model augmentations. However, in this paper we focus on using different dropout masks and experiment mainly with RNNs[2].

## 3 RELATED WORK

### 3.1 RELATION TO EXPECTATION LINEAR DROPOUT (ELD)

Ma et al. (2016) analytically showed that the expected error (over samples) between a model's expected prediction over all dropout masks, and the prediction using the average mask, is upper bounded. Based on this result, they propose to explicitly minimize the difference (we have adapted their regularization to our notations),

$$\mathcal{R}_{ELD}(\mathbf{z}^t; \theta) = \|\mathbb{E}_s\left[\mathbf{p}^t(\mathbf{z}^t, s; \theta)\right] - \mathbf{p}^t(\mathbf{z}^t, \mathbb{E}_s[s]; \theta)\|_2 \quad (4)$$

where $s$ is the dropout mask. However, due to feasibility consideration, they instead propose to use the following regularization in practice,

$$\tilde{\mathcal{R}}_{ELD}(\mathbf{z}^t; \theta) = \mathbb{E}_{s_i}\left[\|\mathbf{p}^t(\mathbf{z}^t, s_i; \theta) - \mathbf{p}^t(\mathbf{z}^t, \mathbb{E}_s[s]; \theta)\|_2^2\right]. \quad (5)$$

Specifically, this is achieved by feed-forwarding the input twice through the network, with and without dropout mask, and minimizing the main network loss (with dropout) along with the regularization term specified above (but without back-propagating the gradients through the network without dropout). The goal of Ma et al. (2016) is to minimize the network loss along with the expected difference between the prediction from individual dropout mask and the prediction from the expected dropout mask. We note that our regularization objective is upper bounded by the expectation-linear dropout regularization as shown below (proof in the appendix).

**Proposition 1.** $\mathcal{R}_{FD}(\mathbf{z}^t; \theta) \le 4\tilde{\mathcal{R}}_{ELD}(\mathbf{z}^t; \theta)$.

This result shows that minimizing the ELD objective indirectly minimizes our regularization term. Finally as indicated above, they apply the target loss only on the network with dropout. In fact, in our own ablation studies (see Section 5) we find that back-propagating target loss through the network (without dropout) makes optimizing the model harder. However, in our setting, simultaneously back-propagating target loss through both networks yields both performance gain as well as convergence gain. We believe convergence is faster for our regularization because network weights are more likely to get target based updates from back-propagation in our case. This is especially true for weight dropout (Wan et al., 2013) since in this case dropped weights do not get updated in the training iteration.

### 3.2 RELATION TO Π-MODEL

Laine & Aila (2016) propose Π-model with the goal of improving performance on classification tasks in the semi-supervised setting. They propose a model similar to ours (considering the equivalent deep feed-forward version of our model) except they apply target loss only on one of the networks and use time-dependent weighting function $\omega(t)$ (while we use constant $\frac{\kappa}{mT}$). The intuition in their case is to leverage unlabeled data by using them to minimize the difference in prediction between the two copies of the network with different dropout masks. Further, they also test their model in the supervised setting but fail to explain the improvements they obtain by using this regularization.

We note that in our case we analytically show that minimizing our regularizer (also used in Π-model) is equivalent to minimizing the variance in the model predictions (Remark 1). Furthermore, we also show the relation of our regularizer to expectation linear dropout (Proposition 1). In Section 5, we

---

[2]The reasons of our focus on RNNs are described in the appendix.

study the effects of target based loss on both networks, which is not used in the Π-model. We find that applying target loss on both the networks leads to significantly faster convergence. Finally, we bring to attention that temporal embedding (another model proposed by Laine & Aila (2016), claimed to be a better version of Π-model for semi-supervised, learning) is intractable in natural language processing applications because storing averaged predictions over all of the time steps would be memory exhaustive (since predictions are usually huge - tens of thousands values). On a final note, we argue that in the supervised case, using a time-dependent weighting function $\omega(t)$ instead of a constant value $\frac{\kappa}{mT}$ is not needed. Since the ground truth labels are known, we have not observed the problem mentioned by Laine & Aila (2016), that the network gets stuck in a degenerate solution when $\omega(t)$ is too large in earlier epochs of training. We note that it is much easier to search for an optimal constant value, which is true in our case, as opposed to tuning the time-dependent function.

Similarity to Π-model makes our method related to other semi-supervised works, mainly Rasmus et al. (2015) and Sajjadi et al. (2016). Since semi-supervised learning is not a primary focus of this paper, we refer to Laine & Aila (2016) for more details.

Another way to address the gap between the train and evaluation mode of dropout is to perform Monte Carlo sampling of masks and average the predictions during evaluation, and this has been used for feed-forward networks. We find that this technique does not work well for RNNs. The details of these experiments can be found in the appendix.

## 4 EXPERIMENTS

### 4.1 LANGUAGE MODELS

In the case of language modeling we test our model[3] on two benchmark datasets – Penn Tree-bank (PTB) dataset (Marcus et al., 1993) and WikiText-2 (WT2) dataset (Merity et al., 2016). We use preprocessing as specified by Mikolov et al. (2010) (for PTB corpus) and Moses tokenizer Koehn et al. (2007) (for the WT2 dataset).

For both datasets we use the AWD-LSTM 3-layer architecture described in Merity et al. (2017a)[4] which we call the baseline model. The number of parameters in the model used for PTB is 24 million as compared to 34 million in the case of WT2 because WT2 has a larger vocabulary size for which we use a larger embedding matrix. Apart from those differences, the architectures are identical. When we use fraternal dropout, we simply add our regularization on top of this baseline model.

**Word level Penn Treebank (PTB)**

Influenced by Melis et al. (2017), our goal here is to make sure that *fraternal dropout* outperforms existing methods not simply because of extensive hyper-parameter grid search but rather due to its regularization effects. Hence, in our experiments we leave a vast majority of hyper-parameters used in the baseline model (Melis et al., 2017) unchanged i.e. embedding and hidden states sizes, gradient clipping value, weight decay and the values used for all dropout layers (dropout on the word vectors, the output between LSTM layers, the output of the final LSTM, and embedding dropout). However, a few changes are necessary:

- the coefficients for AR and TAR needed to be altered because *fraternal dropout* also affects RNNs activation (as explained in Subsection 5.3) – we did not run grid search to obtain the best values but simply deactivated AR and TAR regularizers;
- since *fraternal dropout* needs twice as much memory, *batch size* is halved so the model needs approximately the same amount of memory and hence fits on the same GPU.

The final change in hyper-parameters is to alter the *non-monotone interval* $n$ used in non-monotonically triggered averaged SGD (NT-ASGD) optimizer Polyak & Juditsky (1992); Mandt et al. (2017); Melis et al. (2017). We run a grid search on $n \in \{5, 25, 40, 50, 60\}$ and obtain very

---

[3]Our code is available at `github.com/kondiz/fraternal-dropout`.

[4]We used the official GitHub repository code for this paper `github.com/salesforce/awd-lstm-lm`.

| Model | Parameters | Validation | Test |
|---|---|---|---|
| Zaremba et al. (2014) - LSTM (medium) | 10M | 86.2 | 82.7 |
| Zaremba et al. (2014) - LSTM (large) | 24M | 82.2 | 78.4 |
| Gal & Ghahramani (2016) - Variational LSTM (medium) | 20M | 81.9 | 79.7 |
| Gal & Ghahramani (2016) - Variational LSTM (large) | 66M | 77.9 | 75.2 |
| Inan et al. (2016) - Variational LSTM | 51M | 71.1 | 68.5 |
| Inan et al. (2016) - Variational RHN | 24M | 68.1 | 66.0 |
| Zilly et al. (2016) - Variational RHN | 23M | 67.9 | 65.4 |
| Melis et al. (2017) - 5-layer RHN | 24M | 64.8 | 62.2 |
| Melis et al. (2017) - 4-layer skip connection LSTM | 24M | 60.9 | 58.3 |
| Merity et al. (2017a) - AWD-LSTM 3-layer (baseline) | 24M | 60.0 | 57.3 |
| *Fraternal dropout* + AWD-LSTM 3-layer | 24M | **58.9** | **56.8** |

Table 1: Perplexity on Penn Treebank word level language modeling task.

| Model | Parameters | Validation | Test |
|---|---|---|---|
| Merity et al. (2016) - Variational LSTM + Zoneout | 20M | 108.7 | 100.9 |
| Merity et al. (2016) - Variational LSTM | 20M | 101.7 | 96.3 |
| Inan et al. (2016) - Variational LSTM | 28M | 91.5 | 87.0 |
| Melis et al. (2017) - 5-layer RHN | 24M | 78.1 | 75.6 |
| Melis et al. (2017) - 1-layer LSTM | 24M | 69.3 | 65.9 |
| Melis et al. (2017) - 2-layer skip connection LSTM | 24M | 69.1 | 65.9 |
| Merity et al. (2017a) - AWD-LSTM 3-layer (baseline) | 34M | 68.6 | 65.8 |
| *Fraternal dropout* + AWD-LSTM 3-layer | 34M | **66.8** | **64.1** |

Table 2: Perplexity on WikiText-2 word level language modeling task.

similar results for the largest values (40, 50 and 60) in the candidate set. Hence, our model is trained longer using ordinary SGD optimizer as compared to the baseline model (Melis et al., 2017).

We evaluate our model using the perplexity metric and compare the results that we obtain against the existing state-of-the-art results. The results are reported in Table 1. Our approach achieves the state-of-the-art performance compared with existing benchmarks.

To confirm that the gains are robust to initialization, we run ten experiments for the baseline model with different seeds (without fine-tuning) for PTB dataset to compute confidence intervals. The average best validation perplexity is $60.64 \pm 0.15$ with the minimum value equals $60.33$. The same for test perplexity is $58.32 \pm 0.14$ and $58.05$, respectively. Our score ($59.8$ validation and $58.0$ test perplexity) beats ordinal dropout minimum values.

We also perform experiments using fraternal dropout with a grid search on all the hyper-parameters and find that it leads to further improvements in performance. The details of this experiment can be found in section 5.5.

**Word level WikiText-2 (WT2)**

In the case of WikiText-2 language modeling task, we outperform the current state-of-the-art using the perplexity metric by a significant margin. Due to the lack of computational power, we run a single training procedure for *fraternal dropout* on WT2 dataset because it is larger than PTB. In this experiment, we use the best hyper-parameters found for PTB dataset ($\kappa = 0.1$, non-monotone interval $n = 60$ and halved batch size; the rest of the hyper-parameters are the same as described in Melis et al. (2017) for WT2). The final results are presented in Table 2.

### 4.2 IMAGE CAPTIONING

We also apply *fraternal dropout* on an image captioning task. We use the well-known show and tell model as a baseline[5] (Vinyals et al., 2014). We emphasize that in the image captioning task, the image encoder and sentence decoder architectures are usually learned together. Since we want to focus on the benefits of using *fraternal dropout* in RNNs we use frozen pretrained ResNet-101 (He

---

[5]We used PyTorch implementation with default hyper-parameters from `github.com/ruotianluo/neuraltalk2.pytorch`.

| Model | BLEU-1 | BLEU-2 | BLEU-3 | BLEU-4 |
|---|---|---|---|---|
| Show and Tell Xu et al. (2015) | 66.6 | 46.1 | 32.9 | 24.6 |
| Baseline | 68.8 | 50.8 | 36.1 | 25.6 |
| *Fraternal dropout*, $\kappa = 0.015$ | **69.3** | 51.4 | 36.6 | 26.1 |
| *Fraternal dropout*, $\kappa = 0.005$ | **69.3** | **51.5** | **36.9** | **26.3** |

Table 3: BLEU scores for the Microsoft COCO image captioning task. Using *fraternal dropout* is the only difference between models. The rest of hyper-parameters are the same.

et al., 2015) model as our image encoder. It means that our results are not directly comparable with other state-of-the-art methods, however we report results for the original methods so readers can see that our baseline performs well. The final results are presented in Table 3.

We argue that in this task smaller $\kappa$ values are optimal because the image captioning encoder is given all information in the beginning and hence the variance of consecutive predictions is smaller that in unconditioned natural language processing tasks. *Fraternal dropout* may benefits here mainly due to averaging gradients for different mask and hence updating weights more frequently.

## 5 ABLATION STUDIES

In this section, the goal is to study existing methods closely related to ours – expectation linear dropout Ma et al. (2016), Π-model Laine & Aila (2016) and activity regularization Merity et al. (2017b). All of our experiments for ablation studies, which apply a single layer LSTM, use the same hyper-parameters and model architecture[6] as Melis et al. (2017).

### 5.1 EXPECTATION-LINEAR DROPOUT (ELD)

The relation with expectation-linear dropout Ma et al. (2016) has been discussed in Section 2. Here we perform experiments to study the difference in performance when using the ELD regularization versus our regularization (FD). In addition to ELD, we also study a modification (ELDM) of ELD which applies target loss to both copies of LSTMs in ELD similar to FD (notice in their case they only have dropout on one LSTM). Finally we also evaluate a baseline model without any of these regularizations. The learning dynamics curves are shown in Figure 1. Our regularization performs better in terms of convergence compared with other methods. In terms of generalization, we find that FD is similar to ELD, but baseline and ELDM are much worse. Interestingly, looking at the train and validation curves together, ELDM seems to be suffering from optimization problems.

### 5.2 Π-MODEL

Since Π-model Laine & Aila (2016) is similar to our algorithm (even though it is designed for semi-supervised learning in feed-forward networks), we study the difference in performance with Π-model[7] both qualitatively and quantitatively to establish the advantage of our approach. First, we run both single layer LSTM and 3-layer AWD-LSTM on PTB task to check how their model compares with ours in the case of language modeling. The results are shown in Figure 1 and 2. We find that our model converges significantly faster than Π-model. We believe this happens because we back-propagate the target loss through both networks (in contrast to Π-model) that leads to weights getting updated using target-based gradients more often.

---

[6]We use a batch size of 64, truncated back-propagation with 35 time steps, a constant zero state is provided as the initial state with probability 0.01 (similar to Melis et al. (2017)), SGD with learning rate 30 (no momentum) which is multiplied by 0.1 whenever validation performance does not improve ever during 20 epochs, weight dropout on the hidden to hidden matrix 0.5, dropout every word in a mini-batch with probability 0.1, embedding dropout 0.65, output dropout 0.4 (final value of LSTM), gradient clipping of 0.25, weight decay $1.2 \times 10^{-6}$, input embedding size of 655, the input/output size of LSTM is the same as embedding size (655) and the embedding weights are tied (Inan et al., 2016; Press & Wolf, 2016).

[7]We use a constant function $\omega(t) = \frac{\kappa}{mT}$ as a coefficient for Π-model (similar to our regularization term). Hence, the focus of our experiment is to evaluate the difference in performance when target loss is back-propagated through one of the networks (Π-model) vs. both (ours). Additionally, we find that tuning a function instead of using a constant coefficient is infeasible.

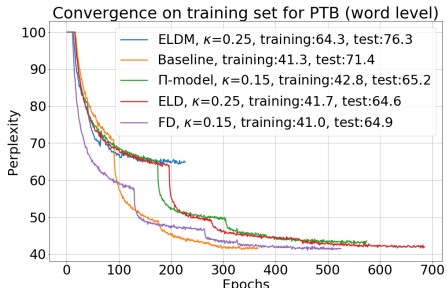
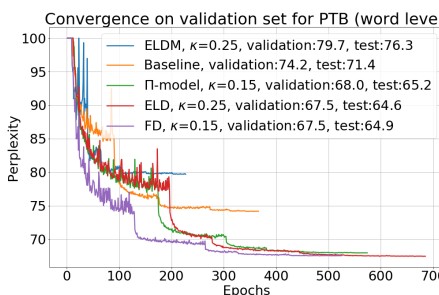

Figure 1: Ablation study: Train (left) and validation (right) perplexity on PTB word level modeling with single layer LSTM (10M parameters). These curves study the learning dynamics of the baseline model, $\Pi$-model, Expectation-linear dropout (ELD), Expectation-linear dropout with modification (ELDM) and *fraternal dropout* (FD, our algorithm). We find that FD converges faster than the regularizers in comparison, and generalizes at par.

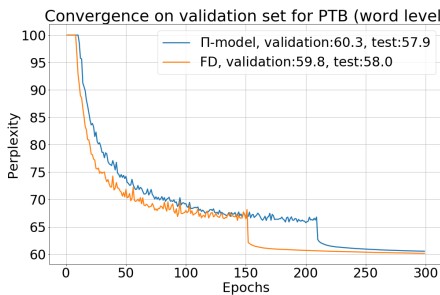
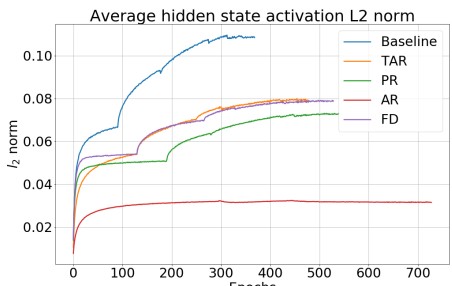

Figure 2: Ablation study: Validation perplexity on PTB word level modeling for $\Pi$-model and *fraternal dropout*. We find that FD converges faster and generalizes at par.

Figure 3: Ablation study: Average hidden state activation is reduced when any of the regularizer described is used. The y-axis is the value of $\frac{1}{d}\|\mathbf{m} \cdot \mathbf{h}^t\|_2^2$.

Even though we designed our algorithm specifically to address problems in RNNs, to have a fair comparison, we compare with $\Pi$-model on a semi-supervised task which is their goal. Specifically, we use the CIFAR-10 dataset that consists of $32 \times 32$ images from 10 classes. Following the usual splits used in semi-supervised learning literature, we use 4 thousand labeled and 41 thousand unlabeled samples for training, 5 thousand labeled samples for validation and 10 thousand labeled samples for test set. We use the original ResNet-56 (He et al., 2015) architecture. We run grid search on $\kappa \in \{0.05, 0.1, 0.15, 0.2\}$, dropout rates in $\{0.05, 0.1, 0.15, 0.2\}$ and leave the rest of the hyperparameters unchanged. We additionally check importance of using unlabeled data. The results are reported in Table 4. We find that our algorithm performs at par with $\Pi$-model. When unlabeled data is not used, *fraternal dropout* provides slightly better results as compared to traditional dropout.

## 5.3 ACTIVITY REGULARIZATION AND TEMPORAL ACTIVITY REGULARIZATION ANALYSIS

The authors of Merity et al. (2017b) study the importance of activity regularization (AR)[8] and temporal activity regularization (TAR) in LSTMs given as,

$$\mathcal{R}_{AR}(\mathbf{z}^t; \theta) = \frac{\alpha}{d}\|\mathbf{h}^t\|_2^2 \tag{6}$$

$$\mathcal{R}_{TAR}(\mathbf{z}^t; \theta) = \frac{\beta}{d}\|\mathbf{h}^t - \mathbf{h}^{t-1}\|_2^2 \tag{7}$$

where $\mathbf{h}^t \in \mathbb{R}^d$ is the LSTM's output activation at time step $t$ (hence depends on both current input $\mathbf{z}^t$ and the model parameters $\theta$). Notice that AR and TAR regularizations are applied on the output of

---

[8]We used $\|\mathbf{m} \cdot \mathbf{h}^t\|_2^2$, where $m$ is the dropout mask, in our actual experiments with AR because it was implemented as such in the original paper's Github repository Merity et al. (2017a).

| Model | Dropout rate | Unlabeled data | Validation | Test |
|---|---|---|---|---|
| Traditional dropout | 0.1 | No | 78.4 ($\pm$ 0.25) | 76.9 ($\pm$ 0.31) |
| No dropout | 0.0 | No | 78.8 ($\pm$ 0.59) | 77.1 ($\pm$ 0.3) |
| *Fraternal dropout* ($\kappa = 0.05$) | 0.05 | No | **79.3** ($\pm$ 0.38) | **77.6** ($\pm$ 0.35) |
| Traditional dropout + $\Pi$-model | 0.1 | Yes | 80.2 ($\pm$ 0.33) | 78.5 ($\pm$ 0.46) |
| *Fraternal dropout* ($\kappa = 0.15$) | 0.1 | Yes | **80.5** ($\pm$ 0.18) | **79.1** ($\pm$ 0.37) |

Table 4: Ablation study: Accuracy on altered (semi-supervised) CIFAR-10 dataset for ResNet-56 based models. We find that our algorithm performs at par with $\Pi$-model. When unlabeled data is not used traditional dropout hurts performance while *fraternal dropout* provides slightly better results. It means that our methods may be beneficial when we lack data and have to use additional regularizing methods.

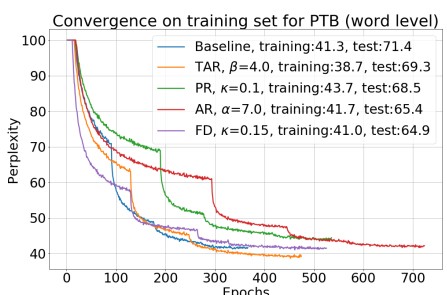
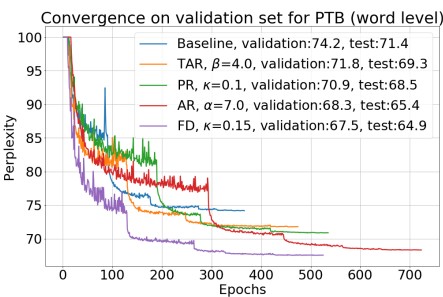

Figure 4: Ablation study: Train (left) and validation (right) perplexity on PTB word level modeling with single layer LSTM (10M parameters). These curves study the learning dynamics of the baseline model, temporal activity regularization (TAR), prediction regularization (PR), activity regularization (AR) and *fraternal dropout* (FD, our algorithm). We find that FD both converges faster and generalizes better than the regularizers in comparison.

the LSTM, while our regularization is applied on the pre-softmax output $\mathbf{p}^t(\mathbf{z}^t, s_i^t; \theta)$ of the LSTM. However, since our regularization can be decomposed as

$$\mathcal{R}_{FD}(\mathbf{z}^t; \theta) = \mathbb{E}_{s_i, s_j}\left[\|\mathbf{p}^t(\mathbf{z}^t, s_i^t; \theta) - \mathbf{p}^t(\mathbf{h}^t, s_j^t; \theta)\|_2^2\right] \tag{8}$$

$$= \mathbb{E}_{s_i, s_j}\left[\|\mathbf{p}^t(\mathbf{z}^t, s_i^t; \theta)\|_2^2 + \|\mathbf{p}^t(\mathbf{z}^t, s_j^t; \theta)\|_2^2 - 2\mathbf{p}^t(\mathbf{z}^t, s_i^t; \theta)^T \mathbf{p}^t(\mathbf{z}^t, s_j^t; \theta)\right] \tag{9}$$

and encapsulates an $\ell^2$ term along with the dot product term, we perform experiments to confirm that the gains in our approach is not due to the $\ell^2$ regularization alone. A similar argument goes for the TAR objective. We run a grid search on $\alpha \in \{1, 2, \ldots, 12\}$, $\beta \in \{1, 2, \ldots, 12\}$, which include the hyper-parameters mentioned in Merity et al. (2017a). For our regularization, we use $\kappa \in \{0.05, 0.1, \ldots, 0.4\}$. Furthermore, we also compare with a regularization (PR) that regularizes $\|\mathbf{p}^t(\mathbf{z}^t, s_i^t; \theta)\|_2^2$ to further rule-out any gains only from $\ell^2$ regularization. Based on this grid search, we pick the best model on the validation set for all the regularizations, and additionally report a baseline model without any of these four mentioned regularizations. The learning dynamics is shown in Figure 4. Our regularization performs better both in terms of convergence and generalization compared with other methods. Average hidden state activation is reduced when any of the regularizer described is applied (see Figure 3).

## 5.4 IMPROVEMENTS USING FINE-TUNING

We confirm that models trained with *fraternal dropout* benefit from the NT-ASGD fine-tuning step (as also used in Merity et al. (2017a)). However, this is a very time-consuming practice and since different hyper-parameters may be used in this additional part of the learning procedure, the probability of obtaining better results due to the extensive grid search is higher. Hence, in our experiments we use the same fine-tuning procedure as implemented in the official repository (even *fraternal dropout* was not used). We present the importance of fine-tuning in Table 5.

| Dropout | Fine-tuning | PTB | | WT2 | |
|---|---|---|---|---|---|
| | | Validation | Test | Validation | Test |
| Traditional | None | 60.7 | 58.8 | 69.1 | 66.0 |
| Traditional | One | 60.0 | 57.3 | 68.6 | 65.8 |
| *Fraternal* | None | 59.8 | 58.0 | 68.3 | 65.3 |
| *Fraternal* | One | 58.9 | 56.8 | **66.8** | **64.1** |
| *Fraternal* | Two | **58.5** | **56.2** | – | – |

Table 5: Ablation study: Importance of fine-tuning for AWD-LSTM 3-layer model. Perplexity for the Penn Treebank and WikiText-2 language modeling tasks.

| Hyper-parameter | Possible values |
|---|---|
| batch size | $[10, 20, 30, 40]$ |
| non-monotone interval | $[5, 10, 20, 40, 60, 100]$ |
| $\kappa$ – FD or ELD strength | $U(0, 0.3)$ |
| weight decay | $U(0.6 \times 10^{-6}, 2.4 \times 10^{-6})$ |

Table 6: Ablation study: Candidate hyper-parameters possible used in the grid search for comparing *fraternal dropout* and *expectation linear dropout*. $U(a, b)$ is the uniform distribution on the interval $[a, b]$. For finite sets, each value is drawn with equal probability.

| Model | Best | Top5 avg | Top10 avg | Beating baseline runs (out of) |
|---|---|---|---|---|
| *Expectation linear dropout* | 59.4 | 60.1 | 60.5 | 6 (208) |
| *Fraternal dropout* | 59.4 | **59.6** | **59.9** | **14** (203) |

Table 7: Ablation study: *Fraternal dropout* and *expectation linear dropout* comparison. Perplexity on the Penn Treebank validation dateset. *Fraternal dropout* is more robust to different hyper-parameters choice as twice as much runs finished performing better than the baseline model (60.7).

## 5.5 FRATERNAL DROPOUT AND EXPECTATION LINEAR DROPOUT COMPARISON

We perform extensive grid search for the baseline model from Subsection 4.1 (an AWD-LSTM 3-layer architecture) trained with either *fraternal dropout* or *expectation linear dropout* regularizations, to further contrast the performance of these two methods. The experiments are run without fine-tuning on the PTB dataset.

In each run, all five dropout rates are randomly altered (they are set to their original value, as in Merity et al. (2017a), multiplied by a value drawn from the uniform distribution on the interval $[0.5, 1.5]$) and the rest of the hyper-parameters are drawn as shown in Table 6. As in Subsection 4.1, AR and TAR regularizers are deactivated.

Together we run more than 400 experiments. The results are presented in Table 7. Both FD and ELD perform better than the baseline model that instead uses AR and TAR regularizers. Hence, we confirm our previous finding (see Subsection 5.3) that both FD and ELD are better. However, as found previously for smaller model in Subsection 5.1, the convergence of FD is faster than that of ELD. Additionally, *fraternal dropout* is more robust to different hyper-parameters choice (more runs performing better than the baseline and better average for top performing runs).

## 6 CONCLUSION

In this paper we propose a simple regularization method for RNNs called *fraternal dropout* that acts as a regularization by reducing the variance in model predictions across different dropout masks. We show that our model achieves state-of-the-art results on benchmark language modeling tasks along with faster convergence. We also analytically study the relationship between our regularization and expectation linear dropout Ma et al. (2016). We perform a number of ablation studies to evaluate our model from different aspects and carefully compare it with related methods both qualitatively and quantitatively.

## ACKNOWLEDGEMENTS

The authors would like to acknowledge the support of the following agencies for research funding and computing support: NSERC, CIFAR, and IVADO. We would like to thank Rosemary Nan Ke and Philippe Lacaille for their thoughts and comments throughout the project. We would also like to thank Stanisław Jastrzębski[†] and Evan Racah[†] for useful discussions.

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

APPENDIX

MONTE CARLO EVALUATION

A well known way to address the gap between the train and evaluation mode of dropout is to perform Monte Carlo sampling of masks and average the predictions during evaluation (MC-eval), and this has been used for feed-forward networks. Since *fraternal dropout* addresses the same problem, we would like to clarify that it is not straight-forward and feasible to apply MC-eval for RNNs. In feed-forward networks, we average the output prediction scores from different masks. However, in the case RNNs (for next step predictions), there is more than one way to perform such evaluation, but each one is problematic. They are as follows:

1. **Online averaging**

Consider that we first make the prediction at time step 1 using different masks by averaging the prediction score. Then we use this output to feed as input to the time step 2, then use different masks at time step 2 to generate the output at time step 2, and so on. But in order to do so, because of the way RNNs work, we also need to feed the previous time hidden state to time step 2. One way would be to average the hidden states over different masks at time step 1. But the hidden space can in general be highly nonlinear, and it is not clear if averaging in this space is a good strategy. This approach is not justified.

Besides, this strategy as a whole is extremely time consuming because we would need to sequentially make predictions with multiple masks at each time step.

2. **Sequence averaging**

Let's consider that we use a different mask each time we want to generate a sequence, and then we average the prediction scores, and compute the argmax (at each time step) to get the actual generated sequence.

In this case, notice it is not guaranteed that the predicted word at time step $t$ due to averaging the predictions would lead to the next word (generated by the same process) if we were to feed the time step $t$ output as input to the time step $t + 1$. For example, with different dropout masks, if the probability of $1^{st}$ time step outputs are: I 40%), he (30%), she (30%), and the probability of the 2nd time step outputs are: am (30%), is (60%), was (10%). Then the averaged prediction score followed by argmax will result in the prediction "I is", but this would be incorrect. A similar concern applies for output predictions varying in temporal length.

Hence, this approach can not be used to generate a sequence (it has to be done by by sampling a mask and generating a single sequence). However, this approach may be used to estimate the probability assigned by the model to a given sequence.

Nonetheless, we run experiments on the PTB dataset using MC-eval (the results are summarized in Table 8). We start with a simple comparison that compares *fraternal dropout* with the averaged mask and the AWD-LSTM 3-layer baseline with a single fixed mask that we call MC1. The MC1 model performs much worse than *fraternal dropout*. Hence, it would be hard to use MC1 model in practice because a single sample is inaccurate. We also check MC-eval for a larger number of models (MC50) (50 models were used since we were not able to fit more models simultaneously on a single GPU). The final results for MC50 are worse than the baseline which uses the averaged mask. For comparison, we also evaluate MC10. Note that no fine-tuning is used for the above experiments.

| Model | Validation | Test |
|---|---|---|
| MC1 | 92.2 ($\pm$ 0.5) | 89.2 ($\pm$ 0.5) |
| MC10 | 66.2 ($\pm$ 0.2) | 63.7 ($\pm$ 0.2) |
| MC50 | 64.4 ($\pm$ 0.1) | 62.1 ($\pm$ 0.1) |
| Baseline (average mask) | 60.7 | 58.8 |
| *Fraternal dropout* | **59.8** | **58.0** |

Table 8: Appendix: Monte Carlo evaluation. Perplexity on Penn Treebank word level language modeling task using Monte Carlo sampling, *fraternal dropout* or average mask.

REASONS FOR FOCUSING ON RNNS

The *fraternal dropout* method is general and may be applied in feed-forward architectures (as shown in Subsection 5.2 for CIFAR-10 semisupervised example). However, we believe that it is more powerful in the case of RNNs because:

1. Variance in prediction accumulates among time steps in RNNs and since we share parameters for all time steps, one may use the same $\kappa$ value at each step. In feed-forward networks the layers usually do not share parameters and hence one may want to use different $\kappa$ values for different layers (which may be hard to tune). The simple way to alleviate this problem is to apply the regularization term on the pre-softmax predictions only (as shown in the paper) or use the same $\kappa$ value for all layers. However, we believe that it may limit possible gains.

2. The best performing RNN architectures (state-of-the-art) usually use some kind of dropout (embedding dropout, word dropout, weight dropout etc.), very often with high dropout rates (even larger than 50% for input word embedding in NLP tasks). However, this is not true for feed-forward networks. For instance, ResNet architectures very often do not use dropout at all (probably because batch normalization is often better to use). It can be seen in the paper (Subsection 5.2, semisupervised CIFAR-10 task) that when unlabeled data is not used the regular dropout hurts performance and using fraternal dropout seems to improve just a little.

3. On a final note, the Monte Carlo sampling (a well known method that adresses the gap beteem the train and evaluation mode of dropout) can not be easily applied for RNNs and *fraternal dropout* may be seen as an alternative.

To conclude, we believe that when the use of dropout benefits in a given architecture, applying *fraternal dropout* should improve performance even more.

As mentioned before, in image recognition tasks, one may experiment with something what we would temporarily dub *fraternal augmentation* (even though dropout is not used, one can use random data augmentation such as random crop or random flip). Hence, one may force a given neural network to have the same predictions for different augmentations.

PROOFS

**Remark 1.** *Let $s_i^t$ and $s_j^t$ be i.i.d. dropout masks and $\mathbf{p}^t(\mathbf{z}^t, s_i^t; \theta) \in \mathbb{R}^m$ be the prediction function as described above. Then,*

$$\mathcal{R}_{FD}(\mathbf{z}^t; \theta) = \mathbb{E}_{s_i^t, s_j^t} \left[ \|\mathbf{p}^t(\mathbf{z}^t, s_i^t; \theta) - \mathbf{p}^t(\mathbf{z}^t, s_j^t; \theta)\|_2^2 \right] = 2 \sum_{q=1}^{m} \mathrm{var}_{s_i^t}(p_q^t(\mathbf{z}^t, s_i^t; \theta)). \quad (10)$$

*Proof.* For simplicity of notation, we omit the time index $t$.

$$\mathcal{R}_{FD}(\mathbf{z}; \theta) = \mathbb{E}_{s_i, s_j} \left[ \|\mathbf{p}(\mathbf{z}, s_i; \theta) - \mathbf{p}(\mathbf{z}, s_j; \theta)\|_2^2 \right] \quad (11)$$

$$= \mathbb{E}_{s_i} \left[ \|\mathbf{p}(\mathbf{z}, s_i; \theta)\|_2^2 \right] + \mathbb{E}_{s_j} \left[ \|\mathbf{p}(\mathbf{z}, s_j; \theta)\|_2^2 \right] \quad (12)$$
$$- 2\mathbb{E}_{s_i, s_j} \left[ \mathbf{p}(\mathbf{z}, s_i; \theta)^T \mathbf{p}(\mathbf{z}, s_j; \theta) \right]$$

$$= 2 \sum_{q=1}^{m} \left( \mathbb{E}_{s_i} \left[ p_q(\mathbf{z}, s_i; \theta)^2 \right] - \mathbb{E}_{s_i, s_j} \left[ p_q(\mathbf{z}, s_i; \theta) p_q(\mathbf{z}, s_j; \theta) \right] \right) \quad (13)$$

$$= 2 \sum_{q=1}^{m} \left( \mathbb{E}_{s_i} \left[ p_q(\mathbf{z}, s_i; \theta)^2 \right] - \mathbb{E}_{s_i} \left[ p_q(\mathbf{z}, s_i; \theta) \right] \mathbb{E}_{s_j} \left[ p_q(\mathbf{z}, s_i; \theta) \right] \right) \quad (14)$$

$$= 2 \sum_{q=1}^{m} \left( \mathbb{E}_{s_i} \left[ p_q(\mathbf{z}, s_i; \theta)^2 \right] - \mathbb{E}_{s_i} \left[ p_q(\mathbf{z}, s_i; \theta) \right]^2 \right) \quad (15)$$

$$= 2 \sum_{q=1}^{m} \mathrm{var}_{s_i}(p_q(\mathbf{z}, s_i; \theta)). \quad (16)$$

$\square$

**Proposition 1.** $\mathcal{R}_{FD}(\mathbf{z}^t; \theta) \leq 4\tilde{\mathcal{R}}_{ELD}(\mathbf{z}^t; \theta)$.

*Proof.* Let $\bar{s} := \mathbb{E}_s[s]$, then

$$\mathcal{R}_t(\mathbf{z}^t) := \mathbb{E}_{s_i^t, s_j^t} \left[ \|\mathbf{p}^t(\mathbf{z}^t, s_i^t; \theta) - \mathbf{p}^t(\mathbf{z}^t, s_j^t; \theta)\|_2^2 \right] \quad (17)$$

$$= \mathbb{E}_{s_i^t, s_j^t} \left[ \|\mathbf{p}^t(\mathbf{z}^t, s_i^t; \theta) - \mathbf{p}^t(\mathbf{z}^t, \bar{s}; \theta) + \mathbf{p}^t(\mathbf{z}^t, \bar{s}; \theta) - \mathbf{p}^t(\mathbf{z}^t, s_j^t; \theta)\|_2^2 \right] \quad (18)$$

$$= 4\mathbb{E}_{s_i^t, s_j^t} \left[ \|\frac{\mathbf{p}^t(\mathbf{z}^t, s_i^t; \theta) - \mathbf{p}^t(\mathbf{z}^t, \bar{s}; \theta)}{2} + \frac{\mathbf{p}^t(\mathbf{z}^t, \bar{s}; \theta) - \mathbf{p}^t(\mathbf{z}^t, s_j^t; \theta)}{2}\|_2^2 \right]. \quad (19)$$

Then using Jensen's inequality,

$$\mathcal{R}_t(\mathbf{z}^t) \leq 4\mathbb{E}_{s_i^t, s_j^t} \left[ \frac{1}{2}\|\mathbf{p}^t(\mathbf{z}^t, s_i^t; \theta) - \mathbf{p}^t(\mathbf{z}^t, \bar{s}; \theta)\|_2^2 + \frac{1}{2}\|\mathbf{p}^t(\mathbf{z}^t, \bar{s}; \theta) - \mathbf{p}^t(\mathbf{z}^t, s_j^t; \theta)\|_2^2 \right] \quad (20)$$

$$= 4\mathbb{E}_{s_i^t} \left[ \|\mathbf{p}^t(\mathbf{z}^t, s_i^t; \theta) - \mathbf{p}^t(\mathbf{z}^t, \bar{s}; \theta)\|_2^2 \right] = 4\tilde{\mathcal{R}}_{ELD}(\mathbf{z}^t; \theta). \quad (21)$$

$\square$

