# OpenReview forum: "Fraternal Dropout"
_ICLR.cc/2018/Conference — Accept (Poster)_

### Official Review · AnonReviewer2 · 2017-11-24
**Improved expectation-linear dropout**

**Rating:** 5
**Confidence:** 4

**Review:**

The authors present Fraternal dropout as an improvement over Expectation-linear dropout (ELD) in terms of convergence and demonstrate the utility of Fraternal dropout on a number of tasks and datasets.

At test time, more often than not, people apply dropout in deterministic mode while at training time masks are sampled randomly. The paper addresses this issue by trying to reduce the gap.

I have 1.5 high level comments:

- Dropout can be applied by averaging results corresponding to randomly sampled masks ('MC eval'). This should not be ignored, and preferrably included in the evaluation.

- It could be made clearer why the proposed regularization would make the aforementioned gap smaller. Intuitively, the bias of the deterministic approximation (compared to the MC eval) should also play a role. It may be worth asking whether the bias changes? A possibility is that MC and deterministic evaluations meet halfway and with fraternal dropout MC eval is worse than without.

Details:

- The notation is confusing: p() looks like a probability distribution, z looks like a latent variable, p^t and l^t have superscripts instead of Y having a subscript, z^t is a function of X. Wouldn't f(X_t) be preferrable to p^t(z_t)?

- The experiments are set up and executed with care, but section 4 could be improved by providings details (as much as in section 5). The results on PTB and Wikitext-2 are really good. However, why not compare to ELD here? Section 5 leads the reader to believe that ELD would be equally good.

- Section 5 could be the most interesting part of the paper. This is where different regularization methods are compared (by the way, this is not "ablation"). It is somewhat unfortunate that due to lack of computational resources the comparisons are made at a single hyperparameter setting.

All in all, the results of section 4 are clearly good, but are they better than those of ELD? Evaluation and interpretation of results in section 5 is made difficult by the omission of the most informative quantity which Fraternal dropout is supposed to be approximating.

---

> ### Author Response · Authors · 2017-12-08
> **AnonReviewer2 (rebuttal)**
>
> Thank you for the detailed comments. Yes, our paper addresses the gap between the train and evaluation mode of dropout. As you mentioned, a well known way to address this gap is to perform MC sampling of masks and average the predictions during evaluation, and this has been used for feed-forward networks. We would like to clarify that it is not straight-forward and feasible to apply this trick for RNNs. In feed-forward networks, we average the output prediction scores from different masks. However, in the case RNNs (for next step predictions), there is more than one way to perform such evaluation, but each one is problematic. These are as follows:
> 1. Let’s consider that we use a different mask each time we want to generate a sequence, and then we average the prediction scores, and compute the argmax (at each time step) to get the actual generated sequence. In this case, notice it is not guaranteed that the predicted word at time step t due to averaging the predictions would lead to the next word (generated by the same process) if we were to feed the time step t output as input to the time step t+1. So this approach is not justified. For example, with different dropout masks, if the probability of 1st time step outputs are: I (40%), he (30%), she (30%), and the probability of the 2nd time step outputs are: am (30%), is (60%), was (10%). Then the averaged prediction score followed by argmax will result in the prediction “I is”, but this would be incorrect. A similar concern applies for output predictions varying in temporal length.
> 2. Consider that we first make the prediction at time step 1 using different masks by averaging the prediction score. Then we use this output to feed as input to the time step 2, then use different masks at time step 2 to generate the output at time step 2, and so on. But in order to do so, because of the way RNNs work, we also need to feed the previous time hidden state to time step 2. One way would be to average the hidden states over different masks at time step 1. But the hidden space can in general be highly nonlinear, and it is not clear if averaging in this space is a good strategy. Besides, this strategy as a whole is more time consuming because we would need to sequentially make predictions with multiple masks at each time step.
>
> We are not sure if we correctly understand the concern of why our method would not make the aforementioned gap smaller. From our perspective, it makes this gap smaller because we show our objective is directly equivalent to the variance in predictions using different dropout masks. Regarding the bias introduced in the objective due to our regularization, we are not sure how to evaluate it. Any suggestions on how to do this would be highly appreciated.
>
> Thank you for your positive comment on our experiments. Regarding evaluating ELD for SOTA PTB and Wikitext-2, we show the comparison of our method with ELD in section 5 and point out that our convergence is faster. We do agree that a comparison for SOTA would also be interesting, and it may be possible that ELD is competitive in terms of final numbers with our method, but since it is slower in terms of convergence and it is computationally intensive to check all methods for SOTA, we only made comparisons with the related works in section 5. But we are currently running SOTA experiments for ELD and if we find the results to be different from what we claim in section 5, we will add them to our paper.
>
> We provided all the hyper-parameter details for section 5 in the footnote because it was a one layer architecture that we used for our ablation studies. For experiments in section 4, we used the exact same architectures and hyper-parameters as provided by Merity et al 2017, so we do not mention them explicitly. Additionally, our code is now publicly available but it was not included in the submitted paper due to double blind submission ( github.com/double-blind-submission/fraternal-dropout ).

---

> > ### Comment · AnonReviewer2 · 2017-12-11
> > **Re: rebuttal**
> >
> > In point 1 of the rebuttal, it is argued that generating a sequence from the model cannot be done by first sampling dropout masks, then generating sequences with each mask and finally averaging the predictions. We agree on this. However, to estimate the probability assigned by the model to a given sequence averaging works fine. In fact, it is the most direct estimate for that probability directly corresponding to loss being optimized. Also, sampling from the model can simply be done by sampling a mask and generating a single sequence.
> >
> > Yes, the alternative in point 2 would be broken.
> >
> > As to the question 'why does fraternal dropout makes the gap smaller?', the question is not whether it does so. The question was how MC eval changes. Does it get worse?
> >
> > Please do add the ELD results for the 'SOTA experiments' to the paper regardless of how they turn out.
> >
> > As to the hyperparameters, comparing the models in section 5 at a single hyperparameter setting has very high variance.

---

> > > ### Author Response · Authors · 2017-12-18
> > > **Further experimental results**
> > >
> > > Thank you for your quick response! As per your suggestion, we are currently running ELD experiments with grid search on the PTB dataset as well as our model with grid search. So far ELD is at par with the baseline (with AR and TAR) and better than the baseline model without AR and TAR regularizations as expected. Due to grid search, we have a slightly better score for fraternal dropout than the one reported in the paper (59.5 validation ppl on validation set, 30 runs), and ELD (60.7 validation ppl on validation set, 34 runs) is worse compared with fraternal dropout. Also, the convergence is better for fraternal dropout on the three layer model, as was also reported in the paper for the single layer model. We will update results (including scores for test set) as more grid runs finish.
> > >
> > > We would like to thank you for your explanation about MC evaluation. We have run experiments on the PTB dataset. We started with a simple comparison that compares fraternal dropout with the averaged mask and the AWD-LSTM baseline with a single fixed mask that we call MC1. The MC1 model achieved on average 92.2 ppl on validation set and 89.2 ppl on test set. Hence, it would be hard to use MC1 model in practice because a single sample is inaccurate. We also checked MC eval for a larger number of models (MC50). We used 50 models since we were not able to fit more on a single GPU we used. The final results for MC50 were on average 64.4 on validation set and 62.1 on test set. Hence, worse than the baseline which uses the averaged mask (60.7 on validation, 58.8 on test). For comparison, the MC10 was also tested, the final results validation and test sets are 66.2 and 63.7, respectively. All experiments were performed used models without fine-tuning.
> > >
> > > Regarding the single hyper-parameter setting, we ran experiments with multiple seeds and found the variance between runs was small. However, only one architecture (number of hidden cells and embedding size) was tested in section 5. We agree that comparison with ELD should be also performed on at least one dataset where SOTA results are claimed and it is why we are currently running more ELD experiments as mentioned before. We picked PTB dataset since it is faster to train model on this data set.

---

### Official Review · AnonReviewer1 · 2017-11-27
**Well written paper on incremental improvement**

**Rating:** 5
**Confidence:** 3

**Review:**

The paper proposes “fraternal dropout”, which passes the same input twice through a model with different dropout masks. The L2 norm of the differences is then used as an additional regulariser. As the authors note, this implicitly minimises the variance of the model under the dropout mask.

The method is well presented and adequately placed within the related work. The text is well written and easy to follow.

I have only two concerns. The first is that the method is rather incremental and I am uncertain how it will stand the test of time and will be adopted.

The second is that of the experimental evaluation. They authors write that a full hyper parameter search was not conducted in the fear of having a more thorough evaluation than the base lines, erroneously reporting superior results.

To me, this is not an acceptable answer. IMHO, the evaluation should be thorough for both the base lines and the proposed method. If authors can get away with a sub standard evaluation because the competing method did, the field might converge to sub standard evaluations overall. This is clearly not in anyones interest. I am open to the author's comments on this, as I understand that spending weeks on tuning a competing method is also not unbiased and work that could be avoided if all software was published.

---

> ### Author Response · Authors · 2017-12-08
> **AnonReviewer1 (rebuttal)**
>
> Thank you for your comments. We would like to bring to the reviewer’s attention that Fraternal dropout is not intended to be a tractable version of the original ELD objective, rather the goal is to actually minimize the variance in predictions. These two are different objectives; hence we believe our method is not an incremental improvement over ELD. Specifically, the original ELD objective targets at enforcing the expected prediction over different masks to be roughly equal to the prediction using the expected mask. This is different from our objective of minimizing the variance across predictions using different masks because it would make each prediction from different mask be similar to the expected prediction over different masks. So our method does not make use of the expected mask anywhere in the computations. We hope this makes the subtle difference between the two methods clear. We establish the relationship between our method and the practical version of ELD (proposed by its authors) because of their similarity.
>
> We completely agree that for a new model/architecture, a thorough grid search should be performed to ensure the evaluations are not sub-standard. However, we propose a regularization method that can be applied on top of existing models. Hence, we used a strong SOTA baseline, and added our single hyperparameter search on top of it. This means that we only tuned a very small subset of all the hyperparameters for our method; the rest were used “as it is” from the baseline model which was heavily tuned by Merity et al. The baseline we used in our paper was the previous SOTA. It is usually the case that hyper-parameter tuning alone does not substantially improve SOTA results on widely-studied benchmarks. Hence the improvement in SOTA we get is a result of our proposed regularization and additional grid search should only make the performance gap higher. Additionally our code is now publically available so everyone can reproduce all our results and check for other hyper-parameters ( github.com/double-blind-submission/fraternal-dropout ).

---

### Official Review · AnonReviewer3 · 2017-11-28
**Presents an interesting idea fairly clearly, but overall very similar to expectation-linear dropout. The analysis is very general and does not really depend on RNN/NN stuff.**

**Rating:** 6
**Confidence:** 3

**Review:**

The proposed method fraternal dropout is a stochastic alternative of the expectation-linear dropout method, where part of the objective is for the dropout mask to have low variance. The first order way to achieve lower variance is to have smaller weights. The second order is by having more evenly spread weights, so there is more concentration around the mean. As a result, it seems that at least part of the effect of explicitly reducing the variance is just stronger weight penalty. The effect of dropout in the first place is the opposite, where variance is introduced deliberately. So I would like to see some comparisons between this method and various dropout rates, and regular weight penalty combinations.

This work is very closely related to expectation linear dropout, except that you are now actually minimizing the variance: 1/2E[ ||f(s) - f(s')|| ] is used instead of E [ ||f(s) - f_bar|| ].  Eq 5 is very close to this, except the f_bar is not quite the mean, but the value with the mean dropout mask. So all the results should be compared with ELD.

I do not think the method is theoretically well-motivated as presented, but the empirical results seem solid.
It is somewhat alarming how the analysis has little to do with the neural networks and how dropout works, let along RNNs, while the strength of the empirical results are all on RNNs.

I feel the ideas interesting and valuable especially in light of strong empirical results, but the authors should do more to clarify what is actually happening.

Minor: why use s_i and s_j, when there is never any reference to i and j? As far as I can tell, i and j serve as constants, more like s_1 and s_2.

---

> ### Author Response · Authors · 2017-12-08
> **AnonReviewer3 (rebuttal)**
>
> We thank you for your constructive comments on our work. We would like to make it clear that our goal was not to design a tractable version of the original ELD objective, but to actually minimize the variance in predictions. We show the relationship with ELD only because it is one of the methods similar to our method.
>
> We will now attempt to clarify that despite the superficial similarity, our method is different from ELD. The original version of ELD proposes to minimize the difference between the expected prediction using different masks and the prediction from the expected mask. Since this term is intractable, the authors of ELD propose a feasible version which has been shown to work well, but the relation between these two variants is unclear. To further illustrate this difference, if we consider predictions made by a linear model: y = m.w.x + b, where m is a Bernoulli mask (with probability of 1 being 0.5), w is a scalar weight, x is a scalar input and b is a scalar bias. Then clearly the original ELD objective is 0 and so the model is not penalized. But notice the practical version of the ELD objective is non-zero, and so this version still penalizes the model. On the other hand, we show that our regularization, which is feasible, directly minimizes the variance in prediction and is upper bounded by the practical version of the ELD objective.
>
> Regarding the comment on enforcing small weights reducing variance, first this weight decay effect from our regularization would directly only happen for the output embedding weight matrix. Secondly, it is unlikely that the weight decay term alone can lead to SOTA results. Thus while this weight decay may in part be helping our model, it cannot be the only factor.
>
> The reviewer seems to be concerned about the proposed method and its analysis being general while we apply it mainly on RNNs. The fraternal dropout method is indeed general and may be applied in feed-forward architectures (as shown in the paper for CIFAR-10 semisupervised example). However, we believe that it is more powerful in the case of RNNs because:
> 1. Variance in prediction accumulates among time steps in RNNs and since we share parameters for all time steps, one may use the same kappa value at each step. In feed-forward networks the layers usually do not share parameters and hence one may want to use different kappa values for different layers (which may be hard to tune). The simple way to alleviate this problem is to apply the regularization term on the pre-softmax predictions only (as shown in the paper) or use the same kappa value for all layers. However, we believe that it may limit possible gains.
> 2. The best performing RNN architectures (state-of-the-art) usually use some kind of dropout (embedding dropout, word dropout, weight dropout etc.), very often with high dropout rates (even larger than 50% for input word embedding in NLP tasks). However, this is not true for feed-forward networks. For instance, ResNet architectures very often do not use dropout at all (probably because batch normalization is often better to use). It can be seen in the fraternal dropout paper (semisupervised CIFAR-10 task) that when unlabeled data is not used the regular dropout hurts performance and using fraternal dropout seems to improve just a little.
>
> Additionally, regarding the comment that our method does not take into consideration how dropout works, notice that dropout can be seen as a form of data augmentation method. From this perspective, our goal in general is to minimize the variance between predictions when different data augmentations are used. So indeed our method is general and does not depend on how dropout works (apart from the fact that the dropout masks are i.i.d.). For instance an analogous regularization technique may be proposed where difference in prediction for two different data augmentation is minimized (for example typically used random flip or crop). But this solution would be applicable for image data only and because it is not general enough was briefly mention it in the updated version of the paper.

---

### Author Response · Authors · 2018-01-05
**Final rebuttal**

In line with our rebuttal, we have made the following changes to our paper to address the points raised by the reviewers:

As suggested by all the reviewers, we performed extensive grid search for an AWD-LSTM 3-layer architecture trained with either fraternal dropout (FD) or expectation linear dropout (ELD) regularizations, to further contrast the performance of these two methods. We have added Subsection 5.5 that summarizes these experiments. In these experiments we confirm that more extensive grid search leads to better results for our approach and FD converges faster than ELD.

As suggested by AnonReviewer2, Monte Carlo evaluation for RNNs with dropout in training mode was studied. We added a subsection in the appendix where our experiment is presented and our discussion with AnonReviewer2 from the rebuttal is summarized. We are thankful for the comments.

As suggested by AnonReviewer3, we described the reasons why we focus on RNNs for applying fraternal dropout as a regularizer (we added additional subsection in the appendix).

We also have made minor changes such as adding missing citations or reordering some parts of our paper.

Finally, we would like to note that the code for language modeling using fraternal dropout is released ( github.com/double-blind-submission/fraternal-dropout ) and the link can be also found in our paper. Hence, our results may be easily replicated (including SOTA results for PTB and WT2 datasets) and/or tested in new configurations.

---

### Public Comment · ~Sungrae_Park1 · 2018-01-19
**released the implementation code  (fraternal dropout)**

The proposed method, fraternal dropout, is the version of self-ensembles (Pi model) for RNNs. The authors proved the fact that the regularization term for self-ensemble is worth for learning RNN models. The results of the paper show incredible performances from the previous state-of-the-art performances on language modeling. I tried to reproduce the performances, and it was easy to get performance with the released codes. Under the same optimizer (Averaged SGD), the performances of the proposed method are converged fast with the better values. I think that the experiments proving that self-ensemble methods also work on RNNs are worth.

---

### Decision · Program_Chairs · 2018-01-29
**ICLR 2018 Conference Acceptance Decision**

**Decision:**

Accept (Poster)

**Comment:**

The paper studies a dropout variant, called fraternal dropout. The paper is somewhat incremental in that the proposed approach is closely related to expectation linear dropout. Having said that, fraternal dropout does improve a state-of-the-art language model on PTB and WikiText2 by ~0.5-1.7 perplexity points. The paper is well-written and appears technically sound.

Some reviewers complain that the authors could have performed a more careful hyperparameter search on the fraternal dropout model. The authors appear to have partly addressed those concerns, which frankly, I don't really agree with either. By doing only a limited hyperparameter optimization, the authors are putting their "own" method at a disadvantage. If anything, the fact that their method gets strong performance despite this disadvantage (compared to very strong baseline models) is an argument in favor of fraternal dropout.